# Evidence in Clinical Studies for the Role of Wall Thickness in Ascending Thoracic Aortic Aneurysms: A Scoping Review

**DOI:** 10.3390/bioengineering10080882

**Published:** 2023-07-25

**Authors:** Gijs P. Debeij, Shaiv Parikh, Tammo Delhaas, Elham Bidar, Koen D. Reesink

**Affiliations:** 1Department of Cardiothoracic Surgery, Heart & Vascular Centre, Maastricht University Medical Centre, 6229 HX Maastricht, The Netherlands; 2Department of Biomedical Engineering, CARIM School for Cardiovascular Diseases, Maastricht University, 6229 ER Maastricht, The Netherlands

**Keywords:** aortopathy, growth and remodeling, measurement techniques

## Abstract

Background: Ascending thoracic aortic aneurysm is a chronic degenerative pathology characterized by dilatation of this segment of the aorta. Clinical guidelines use aortic diameter and growth rate as predictors of rupture and dissection. However, these guidelines neglect the effects of tissue remodeling, which may affect wall thickness. The present study aims to systematically review observational studies to examine to what extent wall thickness is considered and measured in clinical practice. Methods: Using PubMed and Web of Science, studies were identified with data on ascending aortic wall thickness, morphology, aortic diameter, and measurement techniques. Results: 15 included studies report several methods by which wall thickness is measured. No association was observed between wall thickness and aortic diameter across included studies. Wall thickness values appear not materially different between aneurysmatic aortas and non-aneurysmal aortas. Conclusions: The effects on and consequences of wall thickness changes during ATAA formation are ill-defined. Wall thickness values for aneurysmatic aortas can be similar to aortas with normal diameters. Given the existing notion that wall thickness is a determinant of mechanical stress homeostasis, our review exposes a clear need for consistent as well as clinically applicable methods and studies to quantify wall thickness in ascending aortic aneurysm research.

## 1. Introduction

Ascending thoracic aortic aneurysm (ATAA) is a chronic degenerative pathology characterized by dilatation of the ascending thoracic segment of the aorta [1]. The rupture of ATAA leads to a potentially lethal emergency with a pre-hospital mortality of 40%, followed by an increased chance of death of 1% per hour without surgical intervention [2].

ATAA is characterized by a disproportionate degeneration of the media compared to healthy aortas. This eventually results in the remodeling of the aortic extracellular matrix, leading to fewer vascular smooth muscle cells and the degradation of elastin fibers [3]. However, the pathophysiology of aneurysm formation is not entirely understood.

Clinical guidelines use aortic diameter and its growth rate as predictors of rupture and dissection [2], which is the result of aortic tissue ceasing to withstand mechanical stress. These predictors are based on deriving the mechanical stress conditions of the aortic wall under the assumption that wall stress is positively related to diameter. However, according to Laplace’s law, this assumption also requires constant wall thickness over time since wall stress depends on its two main geometrical features, i.e., radius and wall thickness, under a certain pressure condition [4]. Indeed, several observational studies quantifying aortic diameter prior to dissection have demonstrated that most dissections occur in smaller aortic diameters, rendering diameter alone a poor marker for risk stratification [5,6,7]. Admittedly, blood vessels functionally respond to mechanical stress and undergo long-term growth and remodeling via the adaptation of geometry, structure, and material properties [8]. For example, in the event of increased pressure or increased radius leading to increased wall stress, the vessel wall may respond to normalize wall stress by adaptive thickening to normalize wall stress [9]. This kind of adaptation is pursued by the vessel to maintain a mechanical homeostatic state, which is regulated through transmural stress distribution [10]. Unfortunately, studying these conditions in a patient context is cumbersome, causing a lack of evidence for defining the boundaries of these characteristics of growth and remodeling.

Since tissue mechanical stress in the vessel wall is primarily a function of radius and wall thickness, the role of wall thickness in ATAA formation in clinical practice warrants proper attention. 

Therefore, we conducted a scoping review of the literature to examine to what extent wall thickness is considered and measured in clinical research and practice. We followed the PRISMA systematic review methodology to acquire observational clinical studies with explicit data on ascending aortic wall thickness, morphology, and aortic diameter, with a concrete description of the measurement technique.

## 2. Methods

### 2.1. Protocol and Registration

In this study, the Preferred Reporting Items for Systematic Reviews and Meta-Analysis (Prisma) statement [11] is applied. This review was not prospectively registered.

### 2.2. Search Strategy and Study Selection

For the initial search in this review, the PubMed and Web of Science databases have been searched using the following terms: ((aneurysm) AND ((thoracic) OR (ascending))) AND (wall thickness). The search period was set from the earliest available date till August 2021. The last update on the search was on the 3rd of August 2021. 

The articles included had to be written in English, with the primary analysis of tissue data (tissue or image analysis) collected from human ascending aorta focusing on aneurysm. Papers focusing on aortic dissection as well as case reports, review articles and articles that did not present data on overall wall thickness or at least thickness of the intima or media, were excluded from this review. Additionally, papers that focused on validating computational models were excluded, mainly because the focus of these studies is on the validation of the model and not on clinical research.

### 2.3. Study Selection and Data Extraction

The original study idea and the search strategy were conceived by two researchers (G.P.D. and K.D.R.). Screening and selection of papers, based on the title and the abstract, and the inclusion and exclusion criteria were established by one researcher (G.P.D.). After selecting the papers, two researchers (G.P.D. and S.P.) analyzed the full-text papers independently. When available, the following data were extracted from the papers and filled in on a scoresheet: study design, follow-up time, inclusion- and exclusion criteria, size of the study population, technique of wall thickness measurement, the main conclusion regarding wall thickness, overall outcome of the paper, wall thickness data and aortic diameter data. Moreover, the papers were scored on how much they focused on wall thickness in the methods, results, and discussion section (see Appendix A). The two reviewers (G.P.D. and S.P.) discussed their individually filled scoresheets between them. Discrepancies were resolved by reaching an agreement based on a consensus discussion between the two reviewers. If necessary, adjudication was performed through discussion with a third supervisory author (K.D.R.).

### 2.4. Risk of Bias

The risk of bias, and thereby validity, of the included studies in this review, has been independently rated by two reviewers (G.P.D. and S.P.). Any discrepancies while discussing the ratings were resolved by reaching an agreement based on consensus. Using the National Heart, Lung, and Blood Institute, Quality Assessment Tool for Observational Cohort and Cross-Sectional Studies checklist [12], cohort and cross-sectional studies were assessed on their research question, study population, participation rate, in- and exclusion criteria, sample size, exposure measures, outcome measures, blinding, loss to follow-up, confounding variables and overall quality. Case-control studies were assessed by using the National Heart, Lung, and Blood Institute, Quality Assessment of Case-Control Studies checklist [12] rating: research question, study population, sample size, in- and exclusion criteria, case and control definition, exposure prior to condition, exposure definition, blinding of assessors, correction for confounding and overall rating of quality.

### 2.5. Summary of Measures

The primary outcome measure in this review is aortic wall thickness. Wall thickness measures in the included studies were obtained via different measurement techniques. In studies that recorded multiple wall thicknesses taken on the same tissue (e.g., anterior wall, inner- and outer curvature, posterior wall), the average wall thickness was recorded. The average standard deviation of wall thickness in these studies was determined by first calculating the variance of the standard deviations (taking the squares of all standard deviations). Subsequently, the average of the calculated variance was determined, after which the root of the mean variance resulted in the average standard deviation for wall thickness [13]. In this review, we decided to present all wall thickness measurements as median and quartiles since some original papers mentioned median and quartiles because of non-normally distributed wall thickness. For normally distributed measures, the presented mean is equal to the median. The first quartile (Q1) was calculated as *median*—0.67·*standard deviation* [14]. The third quartile (Q3) was calculated as *median* + 0.67·*standard deviation* [14]. For articles that presented an SEM, the standard deviation was calculated by multiplying the SEM by the square root of the sample size. Aortic diameter was defined as a secondary outcome measure. For each study, the average group aortic diameter was obtained via the descriptive characteristics of the included studies or manually calculated in case the study presented the mean diameter separate for each individual participant.

The obtained measurements on wall thickness were visualized using boxplots that show ATAA wall thickness plotted against tissue type and measurement technique. Moreover, a scatter plot has been constructed to visualize the correlation between ATAA diameter and ATAA wall thickness. Since different measurement techniques and tissue types were visualized in this scatterplot, it was not possible to calculate a slope in this scatterplot.

## 3. Results

### 3.1. Study Selection

Using the search strategy, we retrieved 172 articles from PubMed and 143 from the Web of Science. After removing the duplicates (*n* = 83), 232 articles were screened, resulting in 23 eligible studies for full-text analysis. The study selection process is illustrated in Figure 1. After a review of the full articles, eight articles were excluded from analysis because wall thickness or overall wall thickness was not presented (*n* = 6), it was unclear which layer of tissue was measured (*n* = 1) or data on the ascending aorta were not presented (*n* = 1). This resulted in 15 studies, with a total of 1556 ATAA tissue samples, which were included in this review [15,16,17,18,19,20,21,22,23,24,25,26,27,28,29].

### 3.2. Risk of Bias

Results for the quality assessment of the ten included cohort and cross-sectional studies, using the National Heart, Lung, and Blood Institute, Quality Assessment Tool for Observational Cohort and Cross-Sectional Studies checklist, are reported in Table 1. Five papers were deemed of good quality, four papers were of fair quality, and the quality of one paper was rated as poor. It is notable that none of the studies provided a sample size justification. The fact that several papers also have a low sample size might contribute to the risk of bias. Moreover, four out of ten papers did not correct for potential confounding variables, and three out of ten did not clearly describe the study population, which puts them at high risk of bias. On the contrary, all studies did clearly define the outcome and exposure measures and provided a well-described research question/objective.

For the five included case–control studies, using the National Heart, Lung, and Blood Institute, Quality Assessment of Case-Control Studies checklist, the quality assessment is reported in Table 2. The quality of three papers was scored as good, one paper had a fair quality, and one paper was scored as being of poor quality. None of the studies provided a sample size justification which, in combination with the often-low sample size, contributes to the risk of bias. All papers did provide a clear research question and clearly described the in- and exclusion criteria which resulted in a good definition of cases and controls. Two studies [15,19] were rated as having a high risk of bias as they did not properly describe the study population and did not correct for confounding variables. In addition, one of these studies had a high loss to follow-up, while the other study selected cases and controls from different populations. Therefore, caution is needed when interpreting the results of these two studies.

### 3.3. Study Characteristics

A summary of the characteristics of the included studies can be found in Table 3. The selected studies consist of nine cross-sectional studies, five case–control studies and one retrospective cohort study. The amount of collected ATAA tissue samples ranged from 6 to 490 between studies. The mean age range of the study populations was between 43 and 70 (population means). The percentage of female sex among studies varied between 12 and 50. Details for each study are given in Table 3. Primarily, studies reported wall thickness in aneurysm patients with a tricuspid aortic valve. Moreover, four studies also included ATAA patients with bicuspid aortic valves to compare these with ATAA patients with tricuspid aortic valves. Four studies included a control group of healthy aortic tissue samples to compare with aneurysm tissue. One study focused on differences between aneurysm- and dissection tissue. The main exclusion criteria employed in the included articles were dissected aortic tissue and genetic tissue disorders, e.g., Marfan or Loeys-Dietz. In most included studies, ATAA wall thickness was not the primary outcome measurement. ATAA diameter had a prominent role in most included studies since this is the used parameter in current clinical practice.

Overall thickness questionable (see Figure 8 in [21]). The unit of thickness is in cm.

Range for wall thickness was 1.5–2.7 mm (refer Figure 5 in the [23])

### 3.4. Wall Thickness Characteristics

The included studies reported several different wall thickness measurement methods. These methods consisted of an eyepiece micrometer, laser micrometer, line laser triangulation sensors, histomorphometry, digital caliper, electronic caliper, dial gauge, epiaortic ultrasound, measurement trough metal plate, and electronic image analysis. For each measurement method, the wall thickness measurements were further divided into intima-media thickness (IMT) and intima-media-adventitia thickness (IMAT) measures. The ranges of medians of wall thickness measurements for aneurysm patients with different measurement modalities were as follows: medical images 1.3–2.5 mm (IMT), histology 1.3–1.67 mm (IMT) and 1.6–2.5 mm (IMAT), and fresh tissue 1.7–3.1 mm (IMAT). On the other hand, histology IMAT was obtained (range: 1.7–2.6 mm) for NA patients (Figure 2).

In Figure 3, wall thicknesses were plotted against the diameters for every study. The wall thicknesses shown in the figure were the result of a combination of multiple wall thickness measurement methods as described previously. The range of IMT was 1.3–2.5 mm, and IMAT ranged from 1.6 to 3.1 mm. The corresponding ranges of diameters were 40.9–51.0 mm and 45.0–66.0 mm. The measurement ranges included samples from patients with bicuspid and tricuspid aortic valves.

## 4. Discussion

### 4.1. Wall Thickness Data

To our knowledge, this is the first scoping review on the role of aortic wall thickness in ascending thoracic aortic aneurysm formation. At the outset of this review, we investigated in each study the methods for estimating wall thickness (medical imaging, histology or fresh tissue analysis). A varied range in medians of wall thickness measurements was observed, even within measurement techniques. The lowest range that was observed in IMT measurements was on samples measured using histology (~0.5 mm), and a greater range was observed on measurements obtained through medical images (~1 mm). Similarly, the lowest range for IMAT measurements was observed for histological measurements (~1.5 mm), and the measurements on fresh tissue samples ranged greater (~1.5 mm). The reason for the observed variation in ranges could be attributed to the devices used in each study, loading conditions (in vivo, ex vivo), the interpretation by different observers, as well as the differences in population characteristics (Table 3). Additionally, to verify the validity of each measurement method, all included studies were scrutinized for reproducibility and repeatability analysis. It was found that only Cavinato et al. [17] confirmed the repeatability of their method. The method they presented entailed performing wall thickness measurements using laser triangulation sensors on fresh tissue samples (Table 3). They concluded that the method was repeatable. The rest of the included studies did not state the reproducibility or repeatability of the wall thickness measurement methods incorporated in their study. Taken together, there currently exists a lack of consistency across and of quality within studies concerning the measurement of ascending aortic wall thickness. Method and population variability could partially explain the spread in wall thickness we observed.

### 4.2. Interpretation

A key finding is that no noticeable variation in wall thickness with respect to aortic diameters was observed (Figure 3). This finding is opposed to a perceived general characteristic of aortic aneurysms, which is outward hypotrophic remodelling [20,30]. The findings from histopathological studies also describe hyperplasia of VSMCs leading to either thickening or maintenance of thickness instead of thinning of the aortic wall under aneurysmatic conditions [29]. This phenomenon was hypothesized to be a response of VSMCs towards insults to the aortic wall due to hemodynamic forces [29]. Such a hypothesis has been confirmed from animal studies suggesting that the aortic wall undergoes a similar remodelling response when subjected to insult-causing hemodynamic forces [31]. The remodelling response exhibited by the VSMCs “actively” tends to maintain a homeostatic stress state of the vessel wall by regulating the constituents, which leads to variations in the vessel wall thickness given the hemodynamic loading, cell signalling and cell–matrix transduction [8,32]. The variations in the wall thickness ultimately affect the stress state of the tissue and thereby enter a vicious cycle leading to aneurysm formation [33]. The consideration of active cellular contributions to the stress state of the tissue is contrary to considering solely passive vessel material. For example, the wall thickness of a cylinder with a passive (and incompressible) material will always decrease when subjected to increased transmural pressure. Though with some caution, we interpret the lack of a clear correlation between wall thickness and diameter as an indication that the ascending aorta is not a passive elastic but an active structure showing variable degrees of (mal-)adaptation and/or remodelling. For this hypothesis, which is partly at odds with the outward hypotrophic paradigm, there is a pressing need for well-designed studies to better understand how stresses are acting in vivo at the constituent level, which in turn may provide a better understanding of aneurysm formation.

This present review of clinical studies reveals that there is an extensive reliance on ex vivo techniques for determining wall stresses (global and constituent level) and stiffness of the tissue. This is simply due to the fact that the determination of mechanical stresses at the microstructural level of the tissue has been deemed infeasible because of the limited availability of pressure–radius signal, absence of axial force–stretch data, disturbance by involuntary movements of the subject, etc. [34]. Taken together, the interdependency of wall thickness and diameter, particularly in ATAA formation, is interrelated with mechanical stress homeostasis, as regulated at tissue, i.e., the cell–matrix interaction, level.

### 4.3. Limitations

Notwithstanding the role of mechanical stresses in aneurysm formation and the adaptation of the aortic wall, observing the differences in adaptations across populations, with and without genetic disorders, becomes imperative to delve deeper into understanding the interplay between biomechanical and biochemical aspects, ultimately defining the state of aneurysmatic aortic wall. However, we observed an absence of studies presenting wall thickness data on genetic diseases. The reason for this might be that in clinical practice, genetic predisposition is key to diagnosis and follow-up, while the biomechanics of aneurysm formation may then not appear directly relevant clinically. 

Another factor known to determine aortic diameter is the body surface area (BSA) [15]. Several studies reported the use of BSA either to calculate its effect on aortic wall stress [20], normalize ATAA diameter to find failure stress [22], or to calculate the aortic size index (ASI) [27]. Nevertheless, none of the reviewed studies mentioned the effect of BSA on aortic wall thickness.

Yet, another patient factor that is associated with modifications within the extracellular matrix of the aortic wall is age. Age-related modifications may involve collagen cross-linking, elastin fiber re-organization, and VSMC senescence [3,8]. Although studies considered patient populations within an expected 60–65-year age range, the disparity between but also within some studies is substantial (43–75 years; also see Table 3). This may partially explain the (apparent) lack of correlation between wall thickness and diameter we identified. 

At a similar level, the skewed as well as variable sex distribution in the studies we reviewed limits generalization to wider (ATAA) patient populations.

On another note, diabetes status has been conceived to have a protective effect on aneurysm rupture [35]. The effect of diabetes on wall thickness variations also remained unreported in the reviewed studies. Therefore, focused research on the effect of diabetes on wall thickness changes may provide further insight into underlying mechanisms potentially opposing aneurysm growth or rupture/dissection. 

The included papers presented the data on ATAA wall thickness and ATAA diameter in different ways, e.g.: mean ± SD, median ± range. Some papers provided a list of all measurements; moreover, different units of measurement were used between papers. This caused a limitation for our review, as some of the data had to be recalculated from the reported data. Imprecision may have occurred since the calculations in our review were based on rounded numbers from the included papers. Furthermore, the heterogeneity in outcome due to the various wall thickness measurement methods and differences in tissue stress state made it inappropriate to calculate correlations between ATAA wall thickness and ATAA diameter. In addition, calculating correlations stratified for each measurement technique was also not possible because the number of studies per stratum was too small. 

Lastly, our review carries limitations posed by language restrictions as well as the emergent yield of mainly cross-sectional and case–control studies, which by themselves have limited validity [36].

Taken together, the lack of methodological consistency across studies and (inherent) the limitations of individual studies, as well as the limitations of our review, there is a pressing need for prospective, possibly multi-centric studies using the same (single or multiple) wall thickness measurement methods to advance the pathophysiological understanding of ATAA formation in a clinical context.

## 5. Conclusions

In conclusion, multiple measurement techniques and wall thickness measures (IMT or IMAT) were reported across different studies, thereby exposing a lack of a standard for the assessment of aortic wall thickness in ascending aortic aneurysm research. Qualitatively, the variations in wall thickness measurements (aneurysmatic and normal aortas alike) appear to show no clear association with diameter. Given the existing notion that wall thickness is a determinant of mechanical stress homeostasis, our review exposes a clear need for consistent as well as clinically applicable methods and studies to quantify wall thickness in ascending aortic aneurysm research.

## Figures and Tables

**Figure 1 bioengineering-10-00882-f001:**
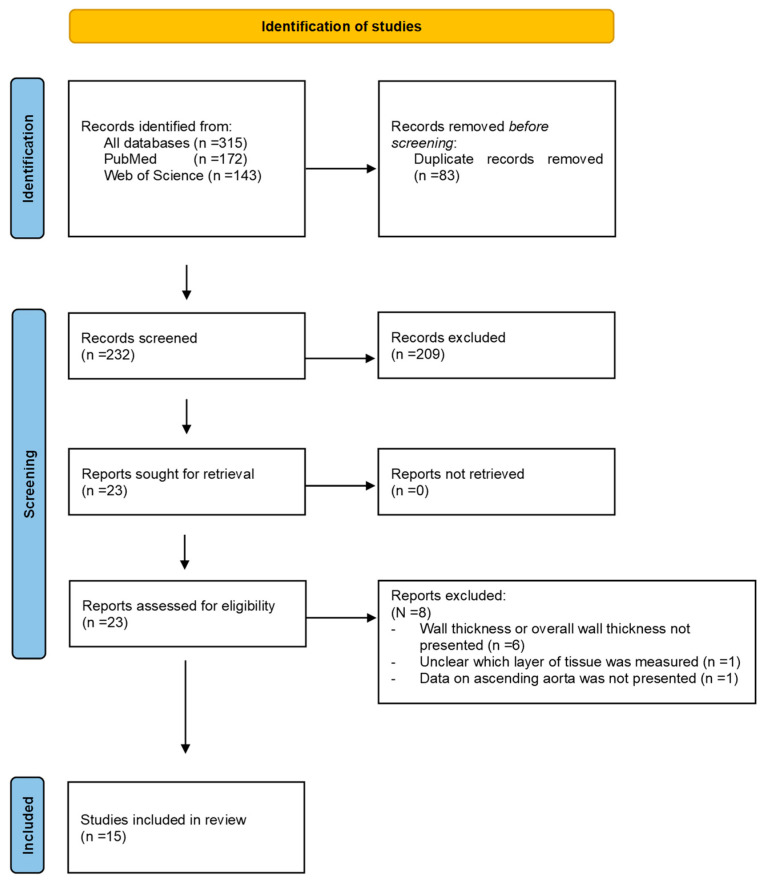
Search and selection procedure.

**Figure 2 bioengineering-10-00882-f002:**
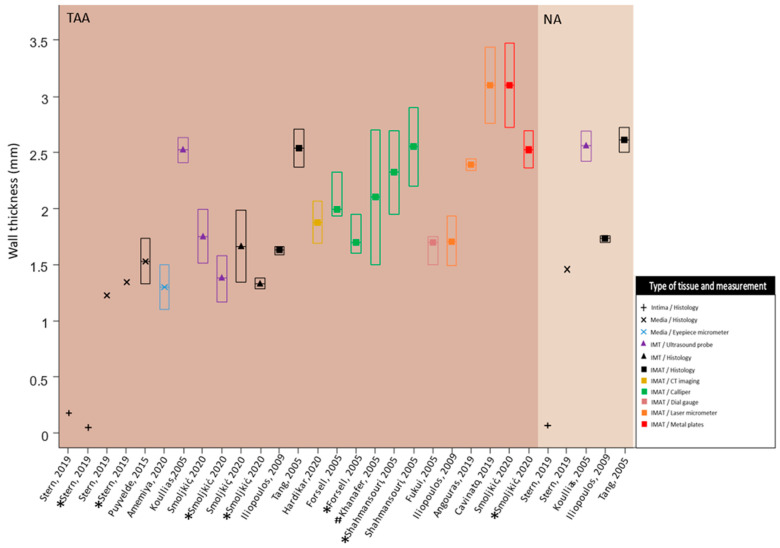
Observing a large range in wall thickness measurements utilizing different measurement techniques. TAA: thoracic aortic aneurysm; NA: non-aneurysm; IMT: intima-media thickness; IMAT: intima-media-adventitia thickness; mm: millimeters; All results are displayed as median [inter-quartile range]. For normally distributed measures the mean is equal to the median. The first quartile (Q1) was calculated as *median*—0.67·*standard deviation* [14]. The third quartile (Q3) was calculated as *median* + 0.67·*standard deviation* [14]; In the TAA group, results for tricuspid and bicuspid aortic valves are displayed; * TAA with bicuspid aortic valve disease; # range of wall thickness is used instead of IQR. Data obtained from [15,16,17,18,19,20,21,22,23,24,25,26,27,28,29].

**Figure 3 bioengineering-10-00882-f003:**
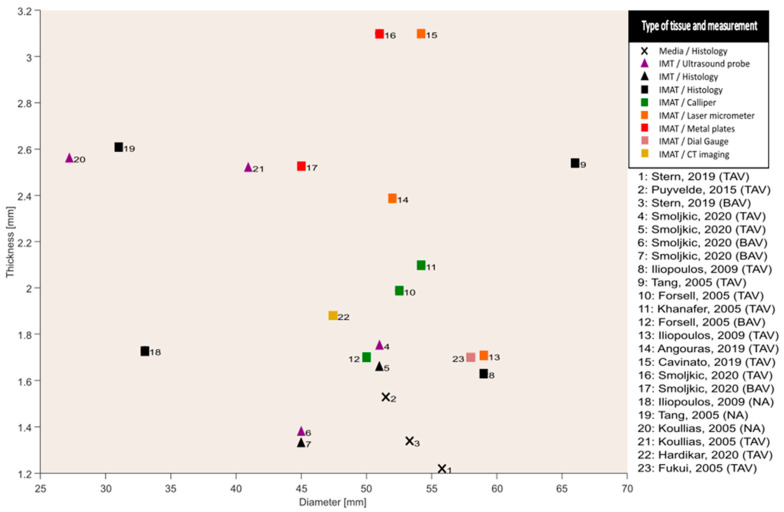
Overall absence of a clear relation between wall thickness and diameter between studies; CT: computed tomography; IMT: intima-media thickness; IMAT: intima-media-adventitia thickness; TAV: tricuspid aortic valve; BAV: bicuspid aortic valve; NA: non-aneurysm; mm: millimeters; median measurements for wall thickness are displayed; average measurements for diameter are displayed. Data obtained from [16,17,18,19,20,21,22,23,24,25,27,28,29].

**Table 1 bioengineering-10-00882-t001:** Outcome of quality assessment for cohort and cross-sectional studies.

Cohort and Cross-Sectional Studies	1. Research Question Clearly Described	2. Study Population Clearly Described	3. At Least 50%? Participation Rate	4. In- and Exclusion Criteria Clearly Described	5. Sample Size Justification Provided	6. Exposure Measured Prior to outcome	7. Timeframe Sufficient to See Association	8. Different Levels of Exposure Measured	9. Exposure Measures Clearly Defined	10. Multiple Exposure Assessments over time	11. Outcome Measures Clearly Defined	12. Outcome Assessors Blinded to Exposure	13. Loss to Follow-Up 20% or Less	14. Correction for Confounding Variables	15. Overall Rating of Quality
Angouras, 2019 [16]	Y	Y	NR	Y	N	Y	Y	Y	Y	NA	Y	NA	Y	N	F
Cavinato, 2019 [17]	Y	Y	NR	Y	N	Y	Y	Y	Y	NA	Y	NA	Y	NA	G
Forsell, 2014 [18]	Y	Y	NR	Y	N	Y	Y	Y	Y	NA	Y	NA	Y	Y	G
Fukui, 2005 [19]	Y	N	NR	Y	N	Y	Y	Y	Y	NA	Y	NA	N	N	P
Hardikar, 2020 [20]	Y	Y	Y	Y	N	Y	Y	Y	Y	N	Y	NA	Y	Y	G
Iliopoulos, 2009 [22]	Y	Y	NR	Y	N	Y	Y	Y	Y	NA	Y	NA	Y	NA	G
Khanafer, 2011 [23]	Y	N	NR	Y	N	Y	Y	Y	Y	NA	Y	NA	Y	N	F
Puyvelde, 2016 [25]	Y	Y	Y	N	N	Y	Y	Y	Y	NA	Y	NR	Y	Y	G
Shahmansouri, 2016 [26]	Y	N	NR	Y	N	Y	Y	Y	Y	NA	Y	Y	CD	Y	F
Smoljkić, 2017 [27]	Y	Y	NR	Y	N	Y	Y	Y	Y	NA	Y	N	NR	N	F

Y: yes; NR: not reported; N: no; NA: not applicable; CD: cannot determine; F: Fair; G: Good; P: Poor.

**Table 2 bioengineering-10-00882-t002:** Outcome of quality assessment for case–control studies.

Case Control Studies	1. Research Question clearly Described	2. Study Population Clearly Described	3. Sample Size Justification	4. Controls from Same Population as Cases	5. In- and Exclusion Criteria Clearly Described	6. Cases/Controls Clearly Defined	7. Randomization in Selecting Participants	8. Concurrent Controls	9. Exposure Occurred prior to Condition	10. Exposure Clearly Defined	11. Assessors of Exposure Blinded for Status	12. Correction for Confounding Variables	13. Overall Rating of Quality
Amemiya, 2020 [15]	Y	N	N	N	Y	Y	NA	N	NA	Y	Y	N	P
Iliopoulos, 2009 [21]	Y	Y	N	Y	Y	Y	NA	N	NA	NA	Y	Y	G
Koullias, 2005 [24]	Y	Y	N	Y	Y	Y	NA	N	Y	Y	N	Y	G
Stern, 2019 [28]	Y	Y	N	Y	Y	Y	NR	N	Y	Y	Y	Y	G
Tang, 2005 [29]	Y	Y	N	NR	Y	Y	NR	N	Y	Y	N	Y	F

Y: Yes; N: No; NA: Not Applicable; NR: Not Reported; P: Poor; G: Good; F: Fair.

**Table 3 bioengineering-10-00882-t003:** Characteristics of included studies.

No.	Study	Study Design	Study Population	Conclusion			
				Method	Result	Discussion	Total Score
1	Amemiya et al., 2020 [15]	Case-control	351 ATAA patient tissue samples with BAV (183, age 56 [46, 66], 21% female), and without BAV (168, age 66 [59, 74], 21% female). In total, 145 TAD patient tissue samples with BAV (8, age 43 [38, 53], 0% females), and without BAV (137, age 65 [55, 76], 37% female).	Media thickness of 5 μm thick paraffin-embedded tissue slices measured using an eyepiece micrometer	Medial wall thickness (TAA) = 1.3 [1.1, 1.5]; Medial wall thickness (TAD) = 1.5 [1.3, 1.8]	Medial degenerative changes (MDC) were observed. Non-BAV is associated with higher MDC scores as compared to BAV aortas. Higher MDC scores are correlated with increased aortic diameters.	1
2	Angouras et al., 2019 [16]	Cross-sectional	17 ATAA patient tissue samples. Age 70 ± 3, 24% female.	Laser micrometer was made (Ls-3100; Keyence Corp, Osaka, Japan). Analysis performed for all four circumferential anatomical locations of the aortic ring.	Anterior = 2.37 ± 0.06; Right lateral = 2.19 ± 0.06; Posterior = 2.33 ± 0.08; Left lateral = 2.66 ± 0.07	Region and not the wall thickness was the cause of reduced delamination strength of aortic tissue. Aortic diameter, which is the main indicator for surgical intervention had no effect on delamination and tensile strength of the tissue.	1
3	Cavinato et al., 2019 [17]	Cross-sectional	12 unruptured ATAA tissue specimens obtained from 19 patients. No mean age available.	High spatial resolution line laser triangulation sensors (optoNCDT 1700BL, Micro-Epsilon Messtechnik GmbH & Co. KG, Germany)	The mean thickness of tissue specimens was 3.10 mm (please refer the paper for local variations in tissue thickness measurements).	No significant correlation found between in vivo rupture pressure and specimen mean or minimum thickness or maximum stresses.	1
4	Forsell et al., 2014 [18]	Cross-sectional	27 tissue specimens from 24 patients undergoing aortic, valve disease, or both surgeries (13 BAV, age 60.4 ± 14.1) (11 TAV, age 59.2 ± 9.4)	Calliper measurements on excised tissue	Increased wall thickness observed in TAV patients (2.0 [1.9, 2.3]) as compared to BAV patients (1.7 [1.6, 2.0])	BAV aneurysmal tissue depicted lower wall thickness as compared to TAV aneurysmal tissue. BAV aneurysmal tissues showed higher strength as compared to TAV aneurysmal tissues.	0
5	Fukui et al., 2005 [19]	Cross-sectional	29 ATAA specimens were obtained from 18 patients undergoing aneurysm replacement surgery. No clinical characteristics mentioned	Measurements on tissue specimens were performed using a dial gauge.	Wall thickness of ascending aortic specimens was 1.7 [1.5, 1.8].	No significant correlations among maximum diameter, wall thickness, and mean infinitesimal strain in the in vivo state.	0
6	Hardikar et al., 2020 [20]	Cross-sectional	72 consecutive patients undergoing aortic surgeries. (58 ATAA, age 61.95 ± 12.1, 16% female) (14 TAD, age 62.40 ± 9.7, 36% female)	Thickness of all four quadrants at mid-ascending level of the aorta was measured using two methods: surgical vernier caliper (after resection of tissue), and from CT images pre-operatively.	Average wall thickness was different for aneurysm (1.88 ± 0.28) and dissection (2.02 ± 0.39). Preferential thinning on the convexity of the aorta was observed with increase in diameter.	Risk of acute aortic events must not be based only on diameter. Aortic wall thickness is an important parameter for risk assessment.	0
7	Iliopoulos et al., 2009 [21]	Case-control	Fresh tissue wall thickness calculation (ATAA *n* = 490; Control *n* = 212). Histomorphometry wall thickness calculation (ATAA *n* = 104; Control *n* = 60). ATAA age 69 ± 2, 35% female. Control age 69 ± 2, 33% female)	Histomorphometry—5 μm thick section images captured using digital camera (Altra20) and analyzed using image analysis software (Image-Pro Plus). Fresh biomechanical testing—non-contacting laser micrometer (LS-3100)	Overall thickness questionable (see Figure 8 in [21]). The unit of thickness is in cm.	Measurements on fresh tissue displayed lower wall thickness in ATAA specimens as compared to controls. No significant differences in overall wall thickness between ATAA and controls from histological analysis. Failure stress and peak elastic modulus correlated negatively with wall thickness (strongly), and ATAA diameter (weakly).	2
8	Iliopoulos et al., 2009 [22]	Cross-sectional	12 ATAA patients, age 69 ± 9, 42% female. In total, 279 specimens of which 271 were submitted for mechanical testing.	Laser beam micrometer (LS3100, Keyence Corp, Osaka, Japan) with resolution of 1 μm. Analysis performed for all four circumferential anatomical locations of the aortic ring.	Anterior = 1.76 ± 0.37; Right lateral = 1.66 ± 0.34; Posterior = 1.66 ± 0.28; Left lateral = 1.79 ± 0.28	No regional variations in wall thickness and failure strain were observed in the ATAA tissue. Negative correlations were found between failure stress and wall thickness in longitudinal and circumferential directions.	1
9	Khanafer et al., 2011 [23]	Cross-sectional	97 aneurysm tissue specimens from 13 patients with age ranging from 39 to 75 years—68 circumferential-oriented specimens (42 greater curvature; 26 lesser curvature) and 29 longitudinal-oriented specimens (16 greater curvature; 13 lesser curvature)	Digital caliper	Range for wall thickness was 1.5–2.7 mm (refer Figure 5 in the [23])	Inverse correlation between peak stress and wall thickness was observed in both—circumferential and longitudinal directions.	2
10	Koullias et al., 2005 [24]	Case-control	20 patients undergoing coronary artery bypass graft surgery, age 64.0 ± 2.61, 30% female. In total, 33 patients undergoing elective aneurysm repair surgery, age 64.8 ± 4.7, 12% female. Patients with a documented diagnosis of Marfan syndrome or evidence of dissection or aortitis of any etiology were excluded.	6- to 15-MHz echocardiographic probe (Phillips model 21390A, Andover, Mass)	Mean value of wall thickness measurement for all non-aneurysmatic tissues was 0.26 ± 0.02 and 0.25 ± 0.02 for aneurysmatic tissues.	Dramatic level of aortic tissue deterioration is observed when the diameter reaches a critical value of 6 cm.	0
11	Puyvelde et al., 2015 [25]	Retrospective cohort	94 ATAA patients, age 62 ± 12.6, 28% female and 87 TAD, age 57.8 ± 15.1, 31% female.	Digitalized histological images processing (software-AxioVision, CarlZeiss Meditec AG, Jena, Germany) of medial wall thickness.	Medial wall thickness ATAA = 1.53 ± 0.29; TAD = 1.50 ± 0.31	Patients with TAD exhibited a significant inverse association between medial wall thickness and aortic diameter, while in ATAA this relation was absent.	2
12	Shahmansouri et al., 2016 [26]	Cross-sectional	Patients with BAV (6 males) and TAV (7 male, 1 female) undergoing aneurysmal repair surgery. Average age of study population is 68 ± 12 years.	Digital caliper	TAV: IC = 2.39 ± 0.43; Anterior = 2.18 ± 0.49; Posterior = 2.72 ± 0.68; OC = 1.992 ± 0.578 BAV: IC = 2.47 ± 0.37; Anterior = 2.56 ± 0.67; Posterior = 2.68 ± 0.49; OC = 2.48 ± 0.50	Regional variation in thickness observed; however, such regional variation for toughness and incremental modulus was not observed. Toughness measure correlates with collagen fiber content in the tissue.	1
13	Smoljkić et al., 2017 [27]	Cross-sectional	6 ATAA patients in total. BAV patients with age 58.25 ± 6. 1 CABG patient aged 60. In total, 2 female patients.	In vivo IMT measurement by images obtained through epiaortic ultrasound probe; Ex vivo IMT measurement on histological samples (at 20 locations and then averaged) through ImageJ; Ex situ IMAT measurement through metal plates and image analysis using MATLAB.	Means of each technique not mentioned. Range for in vivo (IMT)*:* 1–2; ex vivo (IMT): 1.3–2; ex situ (IMAT): 2.2–3.5	Mechanical or geometrical information only cannot provide sufficient information regarding rupture risk. Wall thickness measurements show high variability between patients but also between measurement methods.	1
14	Stern et al., 2019 [28]	Case-control	TAV-ATAA (28, age 65.7 ± 11.3, 43% female); BAV-ATAA (19, age 52.8 ± 14, 5% female); non-aneurysmal (30, age 54.4 ± 12.8, 33% female).	Image acquisitions were performed using Zeiss AxioVision microscope (Carl Zeiss, Oberkochen, Germany) to measure intima and media thickness.	Media thickness significantly reduced in TAV-ATAA patients (1.22) compared to the control group (1.46). The media thickness in patients with a BAV (1.35 mm) displayed no significant differences compared to TAV patients and the control group. Intimal thickness significantly increased in TAV-ATAAs (1.80) compared to BAV-ATAA patients (0.50). No significant intimal thickness difference was observed between TAV-ATAAs and control group (0.75).	BAV-ATAA and the TAV-ATAA are two independent diseases. The TAV-associated aneurysm is characterized by a pronounced aortic wall degeneration suggesting strong wall weakening, whereas BAV aneurysm-associated wall changes are limited to a tendency towards increased calcification.	1
15	Tang et al., 2005 [29]	Case-control	Patients with ATAA (29) age 67.5 ± 14.4, 31% female; patients with non–aneurysmal aortas included CABG (10) + cardiac transplantation (3) + cadaveric organ donors (15). Age of the control group is 60.9 ± 11.1, 21% female.	Wall thickness was determined from elastin van Gieson (EVG)–stained, transverse sections of aortic specimens under magnification using Image 1.62c software (Scion).	Total wall thickness of—non-aneurysmal: 2.61 ± 0.17; smaller aneurysms (15): 2.66 ± 0.15; larger aneurysms (14): 2.54 ± 0.25.	Media becomes thinner as aneurysm develops. Actual mass of media is increased, not decreased. Density of medial VSMCs is preserved in ascending thoracic aortic aneurysms. Increased destruction, most likely via MMP-9, and not decreased synthetic activity, underlies the impaired presence of matrix proteins in the aneurysmal aortic wall.	2

ATAA: ascending thoracic aortic aneurysm; TAD: thoracic aortic dissection; BAV: bicuspid aortic valve; TAV: tricuspid aortic valve; IC: inner curvature; OC: outer curvature; IMT: intima-media thickness; IMAT: intima-media-adventitia thickness. All thickness measurements are in millimeters. Age in years. Results given as Mean ± standard deviation or Median [25th percentile, 75th percentile].

## Data Availability

Data are available upon reasonable request addressed to the corresponding author.

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
