# Peer review of "Evidence in Clinical Studies for the Role of Wall Thickness in Ascending Thoracic Aortic Aneurysms: A Scoping Review"

_bioengineering, 2023, doi:10.3390/bioengineering10080882_

Round 1

Reviewer 1 Report

The study could be really interesting however the risk of bias is hight and there are too many different wall thickness measures.

Only one paper confirms the reproducibility of measures. The finding of the paper as authors say is opposed to a perceived general characteristics of aortic aneurysms. 

Probably a prospective multi center study with standardized measure could be the right way to proceed.

--

Author Response

Dear reviewer,

Thank you for your comments on our manuscript. Please see the attachment for our response to the provided comments and suggestions. We look forward to your reply.

With kind regards,

Gijs Debeij

Reviewer 2 Report

Summary: In this report, the authors describe a systematic meta-analysis attempting to determine how and to what extent aortic wall thickness is considered and measured in the clinical treatment plan for patients with aortic disease. They used PubMed and Web of Science databases to identify suitable indexed peer-reviewed papers that adhered to a strict set of guidelines for comparison purposes. While initially identifying 315 manuscripts, after screening they were left with 15 papers that met their inclusion criteria. From these papers the authors determined that no clear association between wall thickness and aortic diameter could be identified, suggesting that wall thickness values did not differ significantly between aneurysmal and non-aneurysmal aortas. They concluded that these data do not necessarily indicate that measuring aortic wall thickness is irrelevant or unimportant, rather that there is a need to for a deeper understanding of this relationship and its impact on aortic extracellular matrix homeostasis.

Critique:

1.       RESULTS/DISCUSSION (General comments):  If at all possible the authors should include a combined table/figure defining the demographics of the patients utilized in the 15 studies analyzed in this report. It would help the reader to understand whether the 15 papers are comparing patients that are similar or wildly different, providing a better level of confidence that the disparate wall thickness measurements were not due to disparate patient populations.

2.      DISCUSSION: While the author did a great job incorporating several limitations to the data analyzed, patient age should also be considered, given that it is known that aortic compliance and distensibility change as we age. These changes may include both well-described changes within the extracellular matrix, as well as changes in resident cellular function.

3.    DISCUSSION: While the authors very clearly delineate potential reasons as to why the wall thickness data varies may not be sufficient to define a significant relationship with aortic diameter, it may be beneficial to include a paragraph assuming that the results are actually correct. Perhaps wall thickness does not change with respect to aortic diameter. This carries an independent set of significant implications that maybe important to highlight and identify, and may prove to be a significant biomarker in and of itself in terms of rupture risk, homeostatic decompensation, or cellular/structural deficiencies.

Author Response

(The authors gave the same response as above.)
